# A Comparative Study on p- and n-Type Silicon Heterojunction Solar Cells by AFORS-HET

**DOI:** 10.3390/ma15103508

**Published:** 2022-05-13

**Authors:** Wabel Mohammed Alkharasani, Nowshad Amin, Seyed Ahmad Shahahmadi, Ammar Ahmed Alkahtani, Ili Salwani Binti Mohamad, Puvaneswaran Chelvanathan, Tiong Sieh Kiong

**Affiliations:** 1Institute of Sustainable Energy (ISE), Universiti Tenaga Nasional, Jalan IKRAM-UNITEN, Kajang 43000, Malaysia; wabool@yahoo.com (W.M.A.); ammar@uniten.edu.my (A.A.A.); ilisalwani@ieee.org (I.S.B.M.); siehkiong@uniten.edu.my (T.S.K.); 2Engineered Nanosystems Group, School of Science, Aalto University, 00076 Aalto, Finland; 3Solar Energy Research Institute (SERI), Universiti Kebangsaan Malaysia (UKM), Bangi 43600, Malaysia; cpuvaneswaran@ukm.edu.my

**Keywords:** crystalline silicon solar cells, heterojunction, n- and p-types wafers, rear and front-emitters, simulation, AFORS-HET

## Abstract

Despite the increasing trend of n-type silicon wafer utilization in the manufacturing of high-efficiency heterojunction solar cells due to the superior advantages over p-type counterparts, its high manufacturing cost remains to be one of the most crucial factors, which impedes its market share growth with state-of-the-art silicon heterojunction (SHJ) solar cells demonstrating high conversion efficiencies from various configurations, the prospect of using an n-type wafer is debatable from a cost-efficiency point of view. Hence, a systematic comparison between p- and n-type SHJ solar cells was executed in this work using AFORS-HET numerical software. Front and rear-emitter architectures were selected for each type of wafer with ideal (without defects) and non-ideal (with defects) conditions. For ideal conditions, solar cells with p-type wafers and a front-emitter structure resulted in a maximum conversion efficiency of 28%, while n-type wafers demonstrated a maximum efficiency of 26% from the rear-emitter structure. These high-performance devices were possible due to the optimization of the bandgap and electron-affinity for all passivating and doping layers with values ranging from 1.3 to 1.7 eV and 3.9 to 4 eV, respectively. The correlation between the device structure and the type of wafers as demonstrated here will be helpful for the development of both types of solar cells with comparable performance.

## 1. Introduction

The idea of heterojunction silicon solar cells in both amorphous and crystalline configurations was introduced first by Fuhs et al. in 1974 [1], and it later came into reality in 1983 when Hamakawa et al. reported a successful amorphous device with more than 9% power conversion efficiency (PCE) [2]. A decade after, Sanyo Ltd. reported the first heterojunction with an intrinsic thin layer (HIT) using silicon wafers in 1991 reaching an efficiency of 18.1% [3]. 

Since then, several laboratories have actively adopted and explored HIT configurations for further improvement. The primary motivation for this technology comes from the ability to achieve highly efficient cells with low-processing temperature, comparatively low fabrication costs, and optimum surface passivation [4]. Historically, n-type wafers dominated the HIT research and industrial arena [5]. The highest PCE obtained in HIT solar cells used n-type crystalline silicon (c-Si) wafer, whose effectiveness was higher than a p-type wafer [6]. 

This is associated with higher carrier lifetimes, a reduction in light-induced degradation (LID) caused by the formation of boron-oxygen or iron-boron complexes, and not solely relying on the process to enhance the bulk lifetime [7,8,9]. However, recent studies linked with annealing effects in high temperatures within the p-type process, which are capable of reducing LID, and they proved to have successful implementations [9,10,11,12]. Basic problems within the band structure, such as asymmetry offset among the conduction and the valence band (Ec and E_V_), cause the p-type to be less efficient than the n-type as A. Descoeudres reported this in a paper published in 2013 [13]. 

This asymmetry in the band offset triggers a unique recombination reaction in p-type and n-type cells. Large E_V_ offset (∆E_V_) is preferred in n-type wafers, as it minimizes the minority carriers (holes) interfacial recombination probabilities. On the contrary, a small E_C_ offset (∆E_C_) in p-type increases the minority carriers (electrons) interface recombination possibility to occur. Typical n-type wafers are well known to have a better lifetime when compared with p-type wafers. This advantage is recognized from lower defect density and impurities, which enables unwanted recombination. Additionally, n-type Si-doped purely with phosphorus is more resistant to degradation compared to boron-oxygen (B-O) complexes in p-type. 

However, recent investigations indicated that these degradation dynamics could also be found in low-cost n-type wafers as well [14,15]. Currently, p-type wafers are frequently utilized in homojunction cells whether in passivated emitter and rear cell (PERC) design or aluminum back surface field (Al-BSF) solar cells due to the low cost and process simplicity. The cost to produce n-type ingots is considerably higher than for p-type. According to Wang in 2018, it costs 8% more to manufacture n-type wafers [16,17,18]. For now, n-type wafer-based silicon solar cells are less likely to be found in the market compared to p-type. It is still expected for n-type based silicon cells to gain greater market share within the next ten years [19].

It is of great practical importance in manufacturing high-efficiency HIT cells from p-type wafers. Several cells of this form have already shown the possibilities to be made with substantially less PCE than those obtained from wafers of n-type [5,20,21,22,23,24,25]. A variety of studies have already been carried out as mentioned above, which address p- and n-type wafer-based cells, and some are worthy to be mentioned, such as the review De Wolf et al. [6] presented in 2012. They highlighted the fabrication process and its challenges as well as a comparison of different types of structures with results from different groups. 

Another excellent study from Descoeudres et al. [13] was published in 2013. They fabricated two types of cells based on p- and n-type Si wafers. The PCEs of 21.38% for p-type and 22.14% for n-type cells were reported or the recent article in 2019, where 23.76% and 24.21% cells were fabricated, respectively [16]. The current highest HIT solar cell is based on the n-type c-Si wafer with 25.1% by Daisuke et al. in 2015 and Xiaoning et al. in 2020 with some structural optimizations [26,27]. Although silicon solar cells yet dominate the market [19], many studies are shifting towards a variety of thin-film solar cells, including CdTe, CIGS, perovskites, and various polymers. 

Bhattacharya and John [28] presented a new promising methodology that would result in a flexible, high-efficiency thin-film solar cell. However, the presented process is still considered to be complex, and it would cause the fabrication process to be costly. Other groups have focused on the new novel field of perovskites [29,30]. Perovskites proved to have the potential to be the leading technology in the future. However, they are still facing some stability issues [31]. 

Thus, this study focuses on the HIT considering the market acceptances, device stability, production simplicity, and cost-effective procedures with high-efficiency devices [16]. For further improvement of the HIT’s PCE, some factors should be considered and understood carefully before any optimization. The kind of parameters that affect HIT solar cells, such as the structure, thickness, doping concentration, minority carrier lifetime, resistivity, recombination losses, parasitic absorption loss, and defects, usually have a significant effect on the solar cell operation outcome [32,33,34,35,36]. 

Silicon heterojunction studies are usually focused on different aspects of parameter optimization of a-Si as the thin layer for passivation as well as buffer layers or introducing buffer layers [37,38,39,40,41]. In this work, we address the possible effects in PCE or potential limitations in four different HIT structures with defects in three different locations across the bandgap (E_g_) of each layer in a systematic comparison of p- and n-type wafers to understand the effects of defects in various HIT structures using AFORS-HET software [42].

## 2. Methodology

AFORS-HET is considered an essential simulation tool when it comes to heterojunction modeling, especially for silicon solar cells. It utilizes one-dimensional semiconductor equations that are associated with Shockley–Read–Hall statistics, and solves them numerically. As shown in Figure 1, four established HIT silicon solar cells are considered from the literature [13,16]. They are designed based on the two types of wafers, which are n and p-type, the emitter location differs as well, and thus it will be either at the front side (front-emitter cell) or the backside (rear-emitter cell). 

Hence, the potential difference in charge carrier transport across the entire proposed structures can be discussed. As shown, the structures simply contain a c-Si wafer sandwiched between passivating and doped layers. We assumed that the passivating and doping layers have various band gaps and electron affinities. The E_g_ is varied between 1.1 to 2.2 eV, while the electron affinity (χ) is chosen from 3.8 to 4.2 eV. Varying the E_g_ and electron affinity provides a better understanding of the effect of the hydrogen content during fabrication on the cell’s performance. From the experimental point of view, HIT cells are now the best in utilizing passivation contact structures. 

The intrinsic hydrogenated amorphous silicon (a-Si:H) layer is the key component behind the superiority of this structure as it provides outstanding chemical passivation for the absorber. It also can be doped to be p or n-type [43] which could influence an adequately large c-Si surface potential to attain carrier selectivity (electron collection for n-a-Si:H overlayers and hole collection for p-a-Si:H overlayers). The a-Si:H is a direct E_g_ material that depends on the degree of hydrogen incorporation during deposition and could result in E_g_ ranging from 1.5 to 2.0 eV [44,45,46]. As a result, a-Si:H is an excellent material in creating carrier selective contact passivation that works effectively. Furthermore, in a completely formed silicon heterojunction solar cell, there is a remarkable inversion charge at the a-Si:H/c-Si interface, which assists in c-Si surface field-effect passivation. 

Such intrinsic properties could face some offsets due to the extremely high parasitic optical absorption and the difficulty in managing the barrier heterojunction between the a-Si:H/c-Si interface, motivating a continuous search for novel passivation materials. That is another core objective of this study, which is to evaluate the possibility of finding any promising materials in the range of our variation, which are capable of replacing the a-Si:H and provide competitive performance. This variation would likely expound any effect that might occur due to different band offset in absorber or buffer regions on the overall cell performance.

There are a few noticeable differences in the structural parameters, which can be seen in Table 1 that represent the material properties used in the software for each layer. For instance, oxygen defects in p-type c-Si substrate have a density of 1 × 10^10^ cm^−3^ eV^−1^ and are located at 0.55 eV above E_V_ to represent the carrier lifetime of 1 ms. Typically, carrier recombination through defects is presumed to occur by a Shockley–Read–Hall two charge state level. Yet, the silicon dangling bond appears to be the source of interface recombination. At equilibrium, depending on the location of the Fermi level, these defects might be in a neutral, positively charged, or negatively charged state. 

Increased doping level could also result in higher defect density, pinning the Fermi level [6]. In the present simulation, three different locations of defects are chosen and set within the E_g_ where their distributions are depicted in Figure 2. The defects are Gaussian type and acceptor or donor charges with trap density of 1.99 × 10^20^ cm^−3^ eV^−1^ and 0.02 eV energy of distribution. The first location is 0.2 eV above the E_V_, while the second location is exactly at the middle of E_g_ and the final location at 0.2 eV below the E_C_. The ideal device with no defects was assumed to determine the best region from the E_g_ and the χ. Later, the full potential of each structure is investigated.

## 3. Results and Discussion

The highest reported PCE among HIT cells is considered as the benchmark for this study. The obtained electrical results are similar to what has been reported in other works of literature [26,27,47]. The results are divided into separate sections that follow the conditions of the simulation. The first part represents the results of varying the E_g_ and χ of the emitter, BSF, and buffer layers without any defects.

### 3.1. Ideal Case without Defects

Figure 3 depicts the cell efficiency performances of the four structures that are shown in Figure 1. The top left (Figure 3a) represents the performance of the structure in Figure 1a. In this figure, the maximum PCE reached 28% in the E_g_ region from 1.3 to 1.7 eV all over χ axis except when the χ is lesser than 3.85 eV or more than 4.1 eV as the PCE dropped to be lesser than 25% for the first with a dramatic drop for the second with E_g_ more than 1.5 eV. The region below 1.3 eV of the E_g_ is still an active region but with a lower performance with an average of 23%. 

The dead zone is always after 2 eV in the E_g_ or at higher χ (≥4.1) when E_g_ = 1.7 eV. Figure 3b,d are almost identical but with an extra 1% in the highest PCE achieved for Figure 3d. Both of the figures still follow the same trend of Figure 3a where the highest performance is still in the region from E_g_ = 1.3 eV to 1.7 eV. The dead zone is still in the zone after 2 eV in the E_g_ or at higher χ (≥4.1) when E_g_ = 1.7 eV as well. While the region below 1.3 eV of the E_g_ is also performing in a lesser amount in terms of PCE but still higher than 15%. 

However, Figure 3c still follows the same trend as the rest of the graphs where the highest area in the range from E_g_ = 1.3 eV to 1.7 eV, on the other hand, when χ is lesser than 3.85 eV or more than 4.1 eV, the PCE drops. As well as for the dead zone to be after 2 eV for the E_g_ or at higher χ (≥4.1 eV) when E_g_= 1.7 eV. The only difference experienced is at the range below 1.3 eV from the E_g_ as the PCE reach as low as 2%, the reason behind this observation will be discussed later on. The trend shown in Figure 3 indicates that some conditions would provide a better cell performance rather than others and demonstrates the potential for new materials with the same properties to be used as a buffer layer, such as i-μc-Si:H [48]. 

To obtain a better understanding of the results in Figure 3, a deeper analysis of the solar cell performance is required. Therefore, the rest of the cell electrical parameters have been shown in Figure 4 from the structure, which delivers the highest performance that belongs to the structure of Figure 1a. Though there are four different structures, the exact trend from Figure 3 is still the same and applicable for the rest of cell’s performance parameters, and these results are shown in Figure 4 mainly because, on both wafer types, the short-circuit current (J_sc_) measurements are almost identical and reach as high as 43 mA/cm^2^ since the cells’ absorptions of light are similar. 

The ITO and wafer and thicknesses are the same, while p/n-doped a-Si:H layers have identical optical absorption properties [49] and thicknesses. Thus, the outcomes of analyzing one structure would apply to the rest of the structures as they follow the same trend in every aspect of the results obtained.

Figure 4 shows four common performance parameters of a solar cell, open-circuit voltage (V_oc_) in Figure 4a, the J_sc_ in Figure 4b, fill factor (FF) in Figure 4c, and solar cell PCE or η in Figure 4d. In Figure 4a, the V_oc_ has its maximum value of 785 mV at the E_g_ from 1.3 to 2.1 eV, while the χ is less than 3.9 eV. The area below 1.3 eV E_g_ results in V_oc_ around 720 mV irrespective of the χ. The inactive areas are at χ of 3.9 to 4.2 eV and a high value of E_g_ in the vicinity of 2 eV. 

The first look at Figure 4b would remind us of the trend in Figure 3 since the active part is until 1.8 eV in the E_g_ axis but this graph represents the J_sc_ where most of the active part has a value of 43 mA/cm^2^ except when E_g_ is less than 1.2 eV as the value would be in the average of 41 mA/cm^2^. As for the dead zone where the values are almost zero or equal to zero, E_g_ (≥2) has mostly been the region in this section of the study. In Figure 4c, the maximum FF (80%) was calculated in the range of E_g_ =1.3 to 1.6 eV with χ = 3.9 eV onwards and at χ of ≥4.15 eV, the E_g_ region reduces from 1.3 to 1.5 eV. Finally, Figure 4d is as same as Figure 3a detailed above.

It is known that a higher E_g_ value will cause J_sc_ to decrease. However, the higher the E_g_, the higher the V_oc_, based on the equation below.
(1)$$V_{\mathrm{oc}}=\frac{E_{g}}{q}-\frac{\mathrm{kT}}{q}\ln\left(\frac{J_{\mathrm{sc}}}{J_{00}}\right)$$
where J_00_, K, T, and q are the saturation current, Boltzmann constant, temperature, and electron charge, respectively [50]. A minority carrier current (J_n_) is a crucial element, which affects the solar cells’ performance. While having a bias voltage V_b_, the electron minority carrier current is represented by
(2)$$J_{n}=\frac{J_{n0}}{\left(1+v\right)}\times e^{\frac{{\mathrm{qV}}_{b}}{\mathrm{kT}}}$$
where
(3)$$J_{n0}=\mathrm{KT}\frac{\mu_{n1}}{L_{n1}}n_{1}$$
where L_n_, µ_n,_ and n_1_ are the diffusion length, minority-carrier mobility, and concentration, correspondingly. The subscript 1 denotes the depletion region extended in p-type material (Si in our case). The factor ‘v’ shown in Equation (2) states the impact of the ∆E_C_ on the minority carrier electron current and is presented by
(4)$$v=\frac{1}{L_{n1}}\int_{x_{1}}^{x_{2}}\left(\frac{\mu_{n1}N_{c1}}{\mu_{n}N_{c}}\right)e^{-\frac{E_{c1}-E_{c}+q\Psi }{\mathrm{kT}}}\;\mathrm{dx}$$
where x_1_ < 0 to x_2_ > 0 is the depletion region extended from c-Si to a-Si, and Ψ(x) is the potential function. The subscripts 1 and 2 are c-Si and a-Si material, respectively. The abrupt pn heterojunction with applied bias near-zero, solving Equation (4) would be as below.
(5)$$v=\frac{I_{v1}}{L_{n1}}+\frac{1}{L_{n1}}\left(\frac{\mu_{n1}N_{c1}}{\mu_{n}N_{c}}\right)I_{v2}\times e^{-\frac{E_{c1}-E_{c2}}{\mathrm{kT}}}$$
(6)$$\mathrm{where}\;\;{\;I}_{v1}=\int_{x_{1}}^{0}e^{\frac{q\Psi \left(x\right)}{\mathrm{kT}}}\;\mathrm{dx}$$
(7)$$\mathrm{and}\;\;{\;I}_{v2}=\int_{0}^{x_{2}}e^{\frac{q\Psi \left(x\right)}{\mathrm{kT}}}\;\mathrm{dx}$$

The potential function could be assumed to be decoupled by the carrier transport equations and it follows Poisson’s equation as:(8)$$\frac{d^{2}\Psi }{{\mathrm{dx}}^{2}}=-\frac{\rho \left(\Psi \right)}{\epsilon }$$
where ρ is charge distribution and ϵ is the permittivity of the medium. The electrostatic potential Ψ(x) obtained from Equation (8) is used in solving Equations (2)–(7) for a wide E_g_ layer on a narrow E_g_ layer to have [51].
(9)$$J_{n}=\mathrm{kT}\frac{\mu_{n1}N_{c2}n_{i1}^{2}}{I_{v2}N_{c1}N_{A}}e^{-\frac{\left|\Delta E_{c}\right|}{\mathrm{kT}}}\left(e^{\frac{\mathrm{qV}}{\mathrm{kT}}}-1\right)$$

This confirms that the J_n_ decreases significantly for a wide E_g_ layer on a narrow E_g_ layer (a-Si/ c-Si interface) owing to the ∆E_C_ increase [52]. Table 2 shows the results of the ∆E_v_ and the ∆ E_C_ at i-a-Si/c-Si interface. As shown in equation 9, the J_n_ is a function of ∆E_C_, which can result in a lower value at high ∆E_C._ The J_n_ is directly proportional to the J_sc_ where it decreases and decreases at lower and higher J_sc_, respectively. Using equation 1 would explain why the V_oc_ could cover more area in Figure 4a as the V_oc_ increases at higher E_g_. 

The results of Figure 4b line up after comparing the relationship of E_g_ and V_oc_ in equation 1 with ∆E_C_ and J_sc_ in equation 9. As a wide E_g_ will cause higher V_oc_ but lesser J_n_ affecting J_sc_ to be lower as well, ideally, the FF is only a function of V_oc_ [53]. However, the FF might not hinge on the V_oc_ only, the depletion region recombination could play a role as well series resistance losses where those effects are noticeable in Figure 3c as it does not follow the trend of V_oc_ figure. Finally, the PCE = V_oc_ × J_sc_ × FF equation implies that the PCE is a merge of the three output values mentioned previously and established on the importance of V_oc_, J_sc,_ and FF variation in this work.

The effect of ΔE_C_ and ΔE_V_ between the silicon wafer and the intrinsic buffer layer is the main variable that significantly affects the results of Figure 3 specifically the region after E_g_ (≥2 eV) as the PCE drops to zero. Furthermore, corresponding to the type of doping, an increase in E_g_ could be found as upshifting of E_c_ and/or downshifting of E_v_ and proper shifting of E_f_. Variation in the χ moves the entire band structure upward or downward and caused changes in ∆E_C._ For instance, χ of 4.0 eV with E_g_ = 1.5 eV has ΔE_c_ and ΔEv of −0.05 eV and 0.33 eV, respectively, as it is shown in Table 2. 

Band offsets comprehensive analysis of the a-Si:H/c-Si interface done using photoelectron spectroscopy and surface photovoltage measurements, which been published showed that ΔE_C_ and ΔE_V_ do not depend on substrate and film doping, in all cases being approximately equivalent to 0.15 and 0.45 eV, respectively, while fabricating a cell [54,55]. While other data also showed that ΔE_V_ would increase more if the hydrogen content in a-Si:H is being increased, while ΔE_C_ stays constant [56]. These ΔE_C_ offers a mirroring effect for minority carriers, while the ΔE_V_ creates barriers that cause tunneling of the majority carriers. 

The greater ΔE_C_ would cause a potential wall where the minority carrier electrons could be trapped, thus, stopping the process of carrier transport [57]. Overall, at the high values of χ (≥4.1 eV) and E_g_ (≥1.8 eV), results in Table 2 are positive, which suggests a limitation in electron flow from the lower energy E_C_ of the intrinsic layer to the higher energy E_C_ of c-Si. In contrast, lower χ results in negative ΔEc, which indicates a natural electron flow from the higher energy E_C_ of the intrinsic layer to the lower energy E_C_ of c-Si. This also influences the cell performance at low E_g_ values, as electron flow might cause a charge carrier recombination [58].

As mentioned above, Figure 3c does not follow the same trend as the rest of graphs in Figure 3 at the region below 1.3 eV in E_g_. Thus, Figure 5a depicts the band diagram of this solar cell. A lower ΔE_C_ barrier is observed, and this would cause the electrons (minority carriers) to recombine easily at the interface in the p-type case. There are noticeable losses in FF for this structure, while the emitter is located at the back of the device. The FF ranking of various cells is not affected by series resistance losses only but somehow by something intrinsically connected with the device structure. V_oc_ values are somewhat decreased also, but to a lesser extent. The drop in the device PCE could be caused by the illumination reduction on the emitter side of this rear-emitter cell.

The results from Figure 3 represent the best region of E_g_ and χ to be used as a buffer layer and the fact that a high-quality fabricated i-a-Si:H buffer layer in HIT will work as an excellent passivation interface layer. This resulted in shaping the foundation of the other part of this study. The section results can be seen in Figure 6 where we assume that we have an ideal i-a-Si:H layer (E_g_ = 1.7 eV and χ = 3.9 eV) with no defects, which will act as a buffer and passivation layer while keeping the same variation conditions as the previous section regarding the E_g_ and χ variation at the emitter and BSF. 

Therefore, we have the opportunity to study the effects of an ordinary passivation layer on the cell performance while changing E_g_ and χ in the rest of the layers. The four graphs of Figure 6 follow the same trend as the cell performance starts from lower PCE then become stable values and start to decrease again after E_g_ is more than 2.1 eV and χ is higher than 4 eV.

The results of Figure 6 indicate a huge improvement in the overall performance across different bandgaps and affinities. This is mostly due to the improvement in the interface layer after fixing the buffer layer of i-a-Si:H to be of E_g_ of 1.7 eV and χ = 3.9 eV. The band diagram of the system is represented in Figure 5b where ΔE_C_ and ΔE_V_ in these conditions are the same as the fabrication point of view. Using i-a-Si seems to cause a higher built-in voltage at the crystalline absorber while suppressing the interface recombination in the a-Si(emitter)/c-Si heterointerface. As the maximum V_oc_ remains the same in all cells, the J_SC_ increases, with an increase in the overall PCE in structure Figure 1c,d, adding 0.2% more to reach 27.1%. 

The band diagram of Figure 5b indicates that the i-a-Si:H layer would hinder the photon carrier recombination from the n-a-Si:H emitter layer to the p-c-Si base layer, which would enhance J_sc_ and would reflect on the PCE results of the 4 cells. Higher E_g_ and χ would cause a higher value for ΔE_C_. This will suppress the movement of electrons causing the cell performance to decay explaining why the performance dropped again when E_g_ ≥ 2.1 eV and χ ≥ 4 eV. It is clear that the cells still have a trend similar to the previous section, and this indicates that the four structures are useable if they were fabricated specifically.

### 3.2. Non-Ideal State including Defects for All Passivating and Doped Layers

The results of the second case are shown in Figure 7 and Figure 8. In this part of the study, a defect density of 10^−19^ cm^−3^ is introduced to the layers. In the first case, which is represented in Figure 7, all the layers except the Si wafer have a shallow type of defects at 0.2 eV from the E_V_ while keeping the previous conditions of the earlier sections. It is noticeable from Figure 7 that the defects introduced have affected the overall cell performance and reduced the active area in structure n-i-p for p-type Si and p-i-n with n-type Si as illustrated in Figure 7b,c, respectively. 

Figure 7b,c are shown a similar trend as they both have an active area in the region from 1.4 to 1.9 eV in the E_g_ across the different values of χ. While the highest PCE results are when E_g_: 1.5–1.8 eV and 3.9 ≤ χ ≤ 4.1. In these two graphs, the PCE is always below 10% and reaches zero when E_g_ ≤ 1.3 eV for all the χ or when E_g_ ≥ 2 eV. As for the second region, it is expected to remain as a dead zone since it has been a dead zone when the cell is defect free. 

In addition, the region below 1.3 eV has been a low-performance zone in Figure 3c, and the introduction of defects clearly would cause a further reduction in the cell at the same region for Figure 7c results making almost a dead zone for all χ. On the other hand, the graphs of Figure 7a,d are the second patch of trend as the dead zone remains after E_g_ is higher than 2 eV, but a performance decay is observed when E_g_ is below 1.5 eV. The highest PCE region is as same as the other two graph locations mentioned above.

Overall, as the defect density was introduced, the PCE was decreased quite dramatically. This was expected because defect states act as detrimental recombination centers for photo-generated charge carriers [59]. Recombination is lowered in the case where the interface recombining carrier’s concentration is kept low. A large band offset in the minority carrier band grants this condition. 

It is noticeable from Figure 7 that the shallow defects that we introduced had the most impact on the E_g_ values below 1.5 eV through the different structures; however, the unique remark is that the cells with rear emitter endure a severe effect at the same region. This could be a result due to a greater loss in the FF for this type of structure [13]. FF has always been related to series resistance losses; however, the rear emitter structure showed a greater loss, which indicates that this type of structure could experience higher series resistance losses or might be due to the structure layers themselves.

Similar to the earlier section, the second part will address the results while having the ideal passivation layer of i-a-Si, assuming that this layer is perfectly deposited without any defects while adding the shallow defect of the same density stated above to the remaining layers and keeping the E_g_ and χ as variables. All the cell performance simulation results of this segment are depicted in Figure 8. At the first glance at Figure 8, it is clear that it is almost the same as Figure 6 but with some PCE drop within the middle area of Figure 8a and b, which is higher than 1.5 eV of E_g_ but lower than 2.1 eV at χ higher than 4 eV. 

The other two graphs only undergone a decrement of the highest PCE to go from the area started at 1.8 eV to nearly 2.2 eV whilst the χ is lower than 4 eV as it is demonstrated in Figure 6c,d, to shrink and have a small portion located between E_g_ 2–2.1 eV with χ ≥ 4 eV. By comparing Figure 6 and Figure 8, the advantages of using an intrinsic a-Si layer in a heterojunction structure are spotted. The PCE degradation is expected to occur since defects have been added. This layer suppresses the effect of shallow defects near the E_V_ and allows the cell to provide comparable performance as if there were no defects of this type in the E_g_.

### 3.3. Non-Ideal State including Deep Level Defects

As our study advances to the next step of adding defects with the same conditions as the previous part but to a different location as it will be in the middle of the E_g_ in this section, and the results are shown in Figure 9. It is clear at this point that the far-right side of each graph would remain as a dead zone. However, the deep defects showed a greater impact on all of the structures and shrinking the highest PCE regions and the overall active areas. 

Deep defects have higher ionization energy. Thus, these contribute very little to the free charge carriers. Defects with deep levels in the E_g_ are often referred to as traps, recombination centers, or generation centers. Figure 3a,d, which represent the front emitter structure, still shows the same trend, while the rear emitter structures have a similar trend as well. These results indicate that mid-gap defects have a lesser impact on front emitter cells compared to the rear emitter.

Figure 10 depicts the results of applying defects at the middle of the E_g_ while having the ideal i-a-Si:H layer. The n-type silicon wafer still shows similar performance to the cells with defects near the E_V,_ while the p-type cells here illustrate a reduction in the performance at the region above χ 4.05 eV in E_g_ between 1.3 to 1.5 eV and 1.7 to 2 eV. Various recombination behaviors are usually caused by the band offsets asymmetry in n and p-type silicon heterojunction cells. 

The dangling bond defect density goes up when the Fermi level is pushed closer to the E_V_ edge. In n-type wafers, holes are considered as the minority carriers, in that case, a greater valance band offset is beneficial since it reduces the possibility of the interfacial recombination happening. On the other hand, a lesser ∆E_C_ is undesirable for p-type silicon heterojunction cells, since electrons are the minority carriers, as this would result in an increase of the interfacial recombination probability to occur [60].

### 3.4. Non-Ideal State including Shallow Defects

Figure 11 and Figure 12 represent the last part of the study where the defects in this part are located near the Ec, exactly 0.2 eV below the E_c_. The results in this part are very much the same as the results of the prior section addressing the deep defects. It is clear that the deep defects have a similar effect to the one near the Ec for both wafers, which is mainly due to the same reason, which influences the cell performance and causes the decay in certain regions of the graphs. 

The shallow defects near E_v_ showed different behaviors in capturing the carriers and changing the device performance. These shallow defects sometimes could be the impurity elements used as dopants in semiconductors materials, which creates these shallow levels that become ionized at room temperature and provide free carriers. Yet, the region of a-Si:H layer parameter (E_g_ = 1.7 and χ = 3.9 eV) has always been the location of consistent performance. It yielded one of the highest efficiencies in each structure if not the highest in some of the cases mentioned above.

Growing an intrinsic a-Si:H layer on top of crystalline silicon reduces interface recombination efficiently with more symmetry as far as the passivation of different doping types and levels. Moreover, the n-type wafer has proven that it would work under any circumstances if the intrinsic layer is fabricated flawlessly. This indicates that the n-type wafers could perform better than p-type if they have a good passivation layer.

## 4. Conclusions

Numerical simulations of the front and rear-emitter HIT solar cells were investigated using AFORS-HET in this study. Both p- and n-type wafers were compared as a function of various Eg and χ for all passivating and doped layers in both ideal and non-ideal conditions. In ideal conditions, the p-type wafer with a front-emitter structure resulted in 28% of the maximum PCE, while the n-type wafer showed its maximum PCE of 26% from the rear-emitter. These high-performance devices yielded from the optimum regions, while the Eg and χ for all passivating and doping layers were at 1.3 to 1.7 eV and 3.9 to 4 eV, respectively. 

To achieve similar performance at higher χ (e.g., 4 to 4.2 eV), the Eg should be between 1.3 and 1.5 eV. This trend was similar in rear and front-emitter structures using p- and n-type wafers; however, the performance decreased slightly. We found that a-Si:H buffer layers may couple well with doping layers with the Eg from 1.3 to 2 eV and 1.8 to 2 eV for front and rear-emitter structures regardless of the type of wafers. However, the front and rear-emitter structures still provided higher efficiencies for p- and n-types. 

In non-ideal conditions, the effects of defects on passivating and doping layers led to the optimum regions being limited. We found that p-type devices produced a similar performance for both front and rear-emitter structures assuming that the levels of the defects remained at the middle or close to the Ec for all passivating and doped layers. This observation validates the advantages of using p-type wafers in this scenario. In contrast, the insertion of a-Si:H as passivating layers brings significant advantages to n-type wafers by widening the optimum region. Finally, these observations were successfully correlated with the band offsets and the illumination reduction by the a-Si:H layer at the emitter side.

## Figures and Tables

**Figure 1 materials-15-03508-f001:**
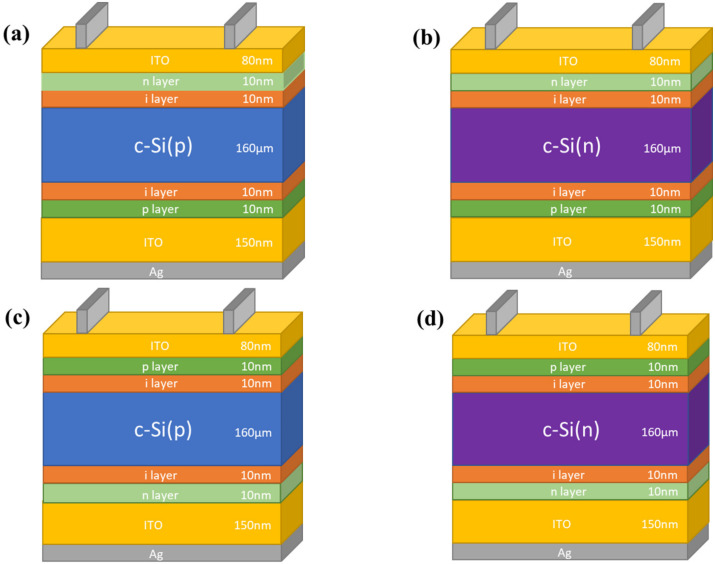
Schematic structure of HIT solar cells. (**a**) p-Type wafer with n-i-p structure (front-emitter cell). (**b**) n-Type wafer with n-i-p structure (front-emitter cell). (**c**) p-Type wafer with p-i-n structure (rear-emitter cell). (**d**) n-Type wafer with p-i-n structure (rear-emitter cell).

**Figure 2 materials-15-03508-f002:**
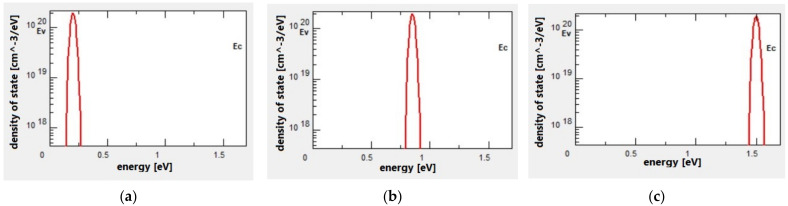
(**a**–**c**) Defect state distributions in a-Si layers.

**Figure 3 materials-15-03508-f003:**
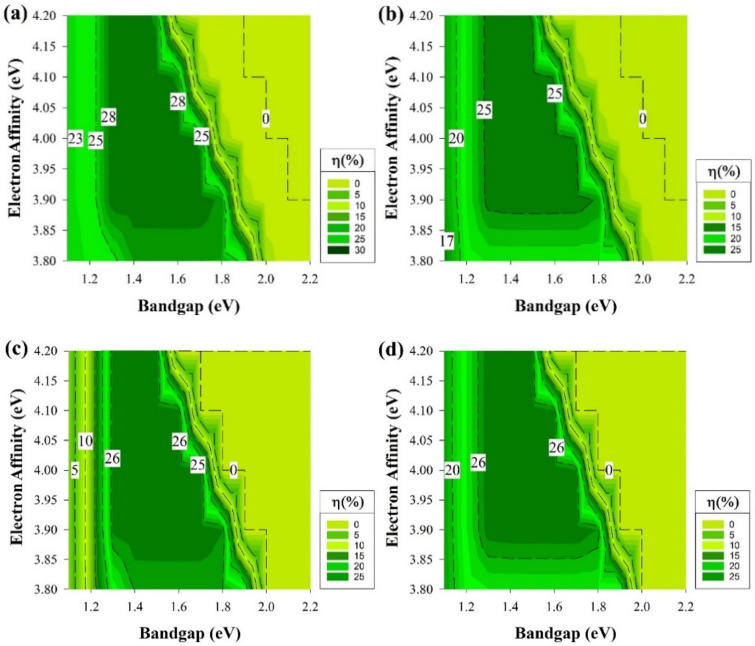
The efficiency of four proposed structures as a function of E_g_ and χ for all passivating and doped layers without defects. (**a**) Contour graphs of the n-i-p structure with p-type wafer base. (**b**) Contour graphs of the n-i-p structure with n-type wafer base. (**c**) Contour graphs of the p-i-n structure with the p-type wafer. (**d**) Contour graphs of the p-i-n structure with n-type wafer base.

**Figure 4 materials-15-03508-f004:**
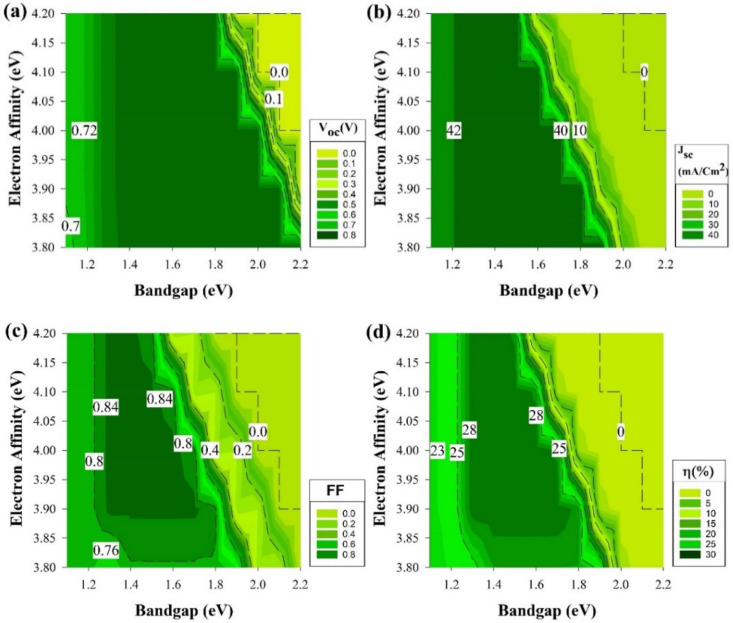
Performance parameters of the structure in Figure 1a.

**Figure 5 materials-15-03508-f005:**
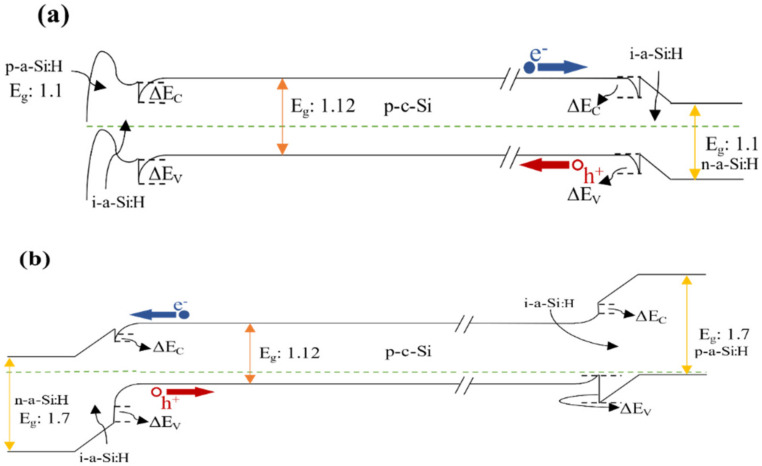
(**a**) The corresponding band diagram of structure p-i-n with p-type wafer when E_g_ = 1.1 and χ = 4. (**b**) The corresponding band diagram of structure n-i-p when E_g_ = 1.7 and χ = 3.9.

**Figure 6 materials-15-03508-f006:**
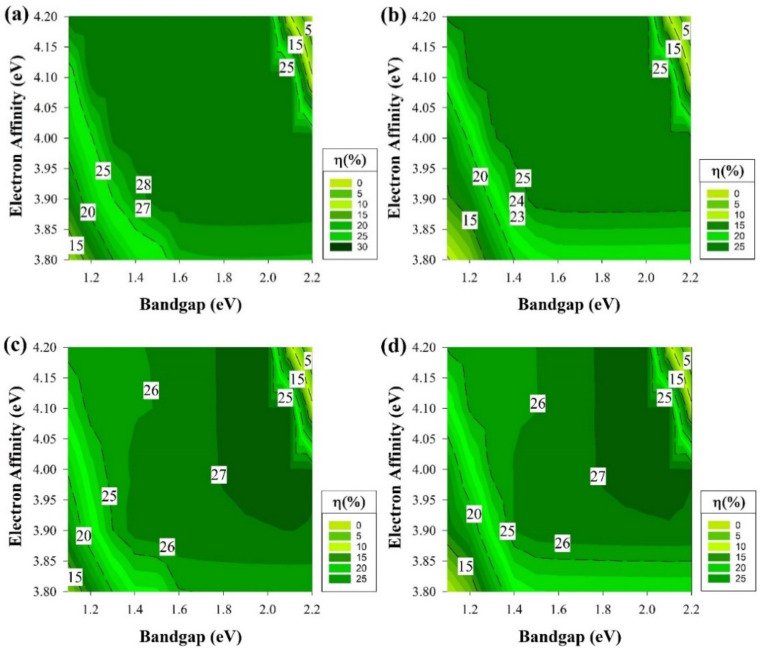
The efficiency of four proposed structures, including a-Si:H passivating layers as a function of Eg and χ for all doped layers without defects. (**a**) Contour graphs of the n-i-p structure with p-type wafer base. (**b**) Contour graphs of the n-i-p structure with n-type wafer base. (**c**) Contour graphs of the p-i-n structure with the p-type wafer. (**d**) Contour graphs of the p-i-n structure with n-type wafer base.

**Figure 7 materials-15-03508-f007:**
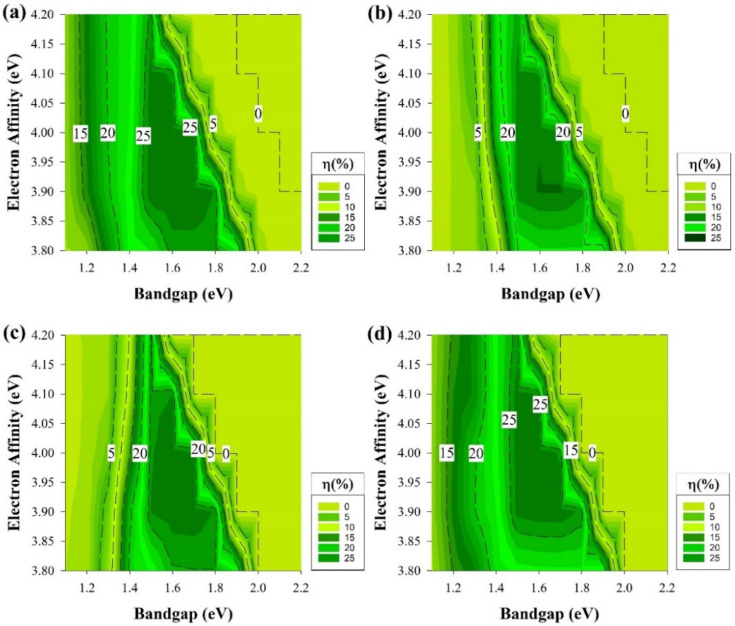
The efficiency of four proposed structures as a function of Eg and χ for all passivating and doped layers, including defects at the 0.2 eV level above the E_V_. (**a**) Contour graphs of the n-i-p structure with p-type wafer base. (**b**) Contour graphs of the n-i-p structure with n-type wafer base. (**c**) Contour graphs of the p-i-n structure with the p-type wafer. (**d**) Contour graphs of the p-i-n structure with n-type wafer base.

**Figure 8 materials-15-03508-f008:**
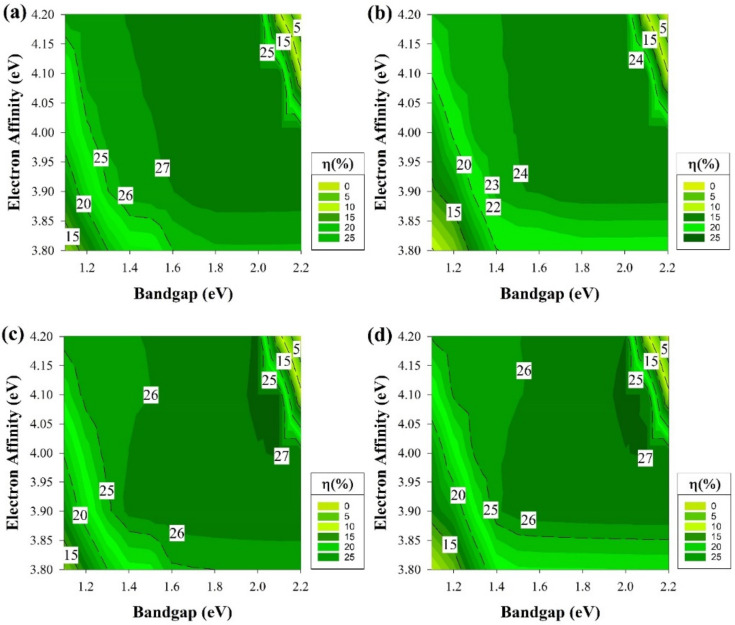
The efficiency of four proposed structures, including a-Si:H passivating layers as a function of E_g_ and χ for doped layers with defects at the 0.2 eV level above E_V_. (**a**) Contour graphs of the n-i-p structure with p-type wafer base. (**b**) Contour graphs of the n-i-p structure with n-type wafer base. (**c**) Contour graphs of the p-i-n structure with the p-type wafer. (**d**) Contour graphs of the p-i-n structure with n-type wafer base.

**Figure 9 materials-15-03508-f009:**
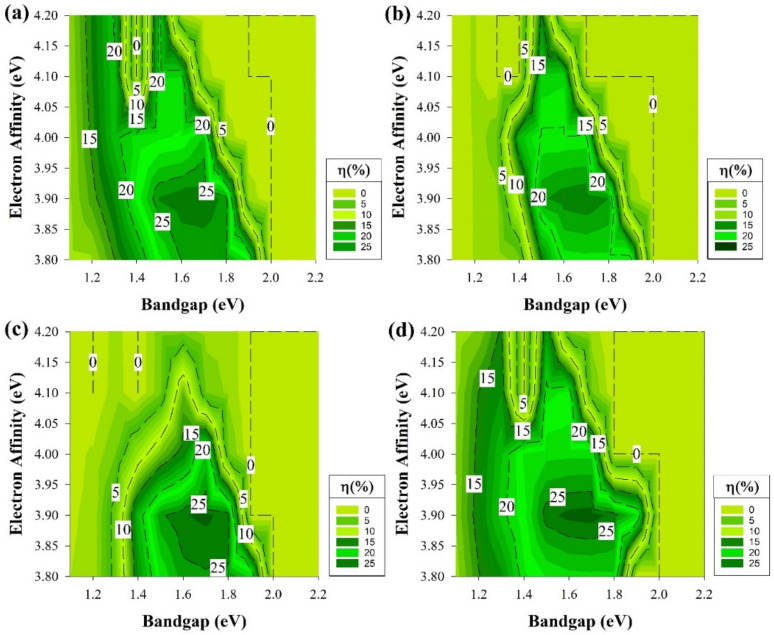
The efficiency of four proposed structures as a function of E_g_ and χ for all passivating and doped layers, including defects at the middle of E_g_ (**a**) Contour graphs of the n-i-p structure with p-type wafer base. (**b**) Contour graphs of the n-i-p structure with n-type wafer base. (**c**) Contour graphs of the p-i-n structure with the p-type wafer. (**d**) Contour graphs of the p-i-n structure with n-type wafer base.

**Figure 10 materials-15-03508-f010:**
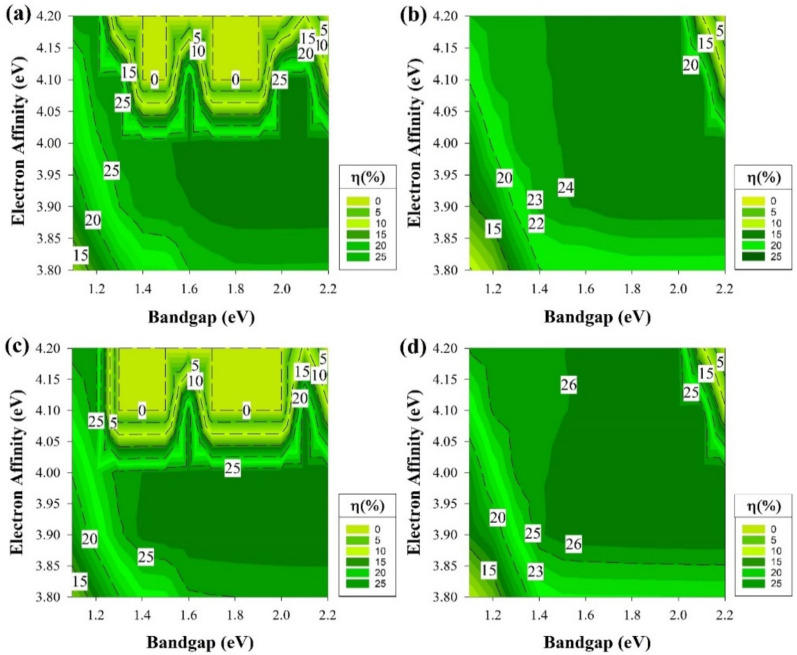
The efficiency of four proposed structures, including a-Si:H passivating layers as a function of Eg and χ for doped layers with defects at the middle of E_g_ (**a**) Contour graphs of the n-i-p structure with p-type wafer base. (**b**) Contour graphs of the n-i-p structure with n-type wafer base. (**c**) Contour graphs of the p-i-n structure with the p-type wafer. (**d**) Contour graphs of the p-i-n structure with n-type wafer base.

**Figure 11 materials-15-03508-f011:**
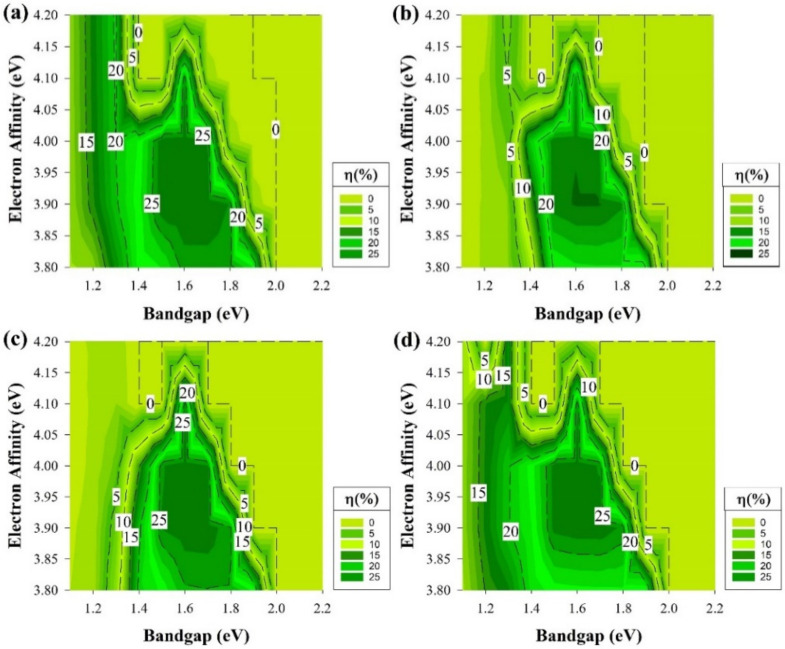
The efficiency of four proposed structures as a function of Eg and χ for all passivating and doped layers, including defects at the 0.2 eV level below Ec. (**a**) Contour graphs of the n-i-p structure with p-type wafer base. (**b**) Contour graphs of the n-i-p structure with n-type wafer base. (**c**) Contour graphs of the p-i-n structure with the p-type wafer. (**d**) Contour graphs of the p-i-n structure with n-type wafer base. (**a**–**d**) All graphs show the efficiency performance while varying the Eg and χ.

**Figure 12 materials-15-03508-f012:**
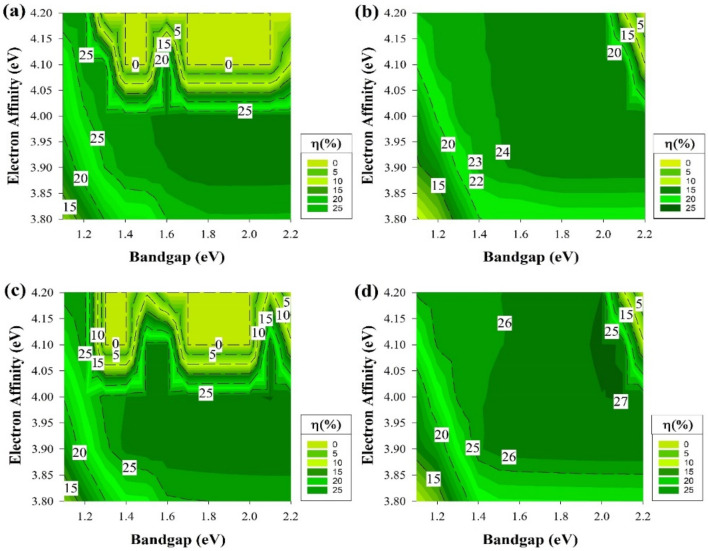
The efficiency of four proposed structures, including a-Si:H passivating layers as a function of Eg and χ for doped layers with defects at the 0.2 eV level below Ec. (**a**) Contour graphs of the n-i-p structure with p-type wafer base. (**b**) Contour graphs of the n-i-p structure with n-type wafer base. (**c**) Contour graphs of the p-i-n structure with the p-type wafer. (**d**) Contour graphs of the p-i-n structure with n-type wafer base.

**Table 1 materials-15-03508-t001:** Parameters used in the simulation.

Parameters	c-p-Si	c-n-Si	i-Layer	p-Layer	n-Layer
Layer thickness (nm)	16 × 10^4^	16 × 10^4^	10	10	10
Dielectric constant	11.9	11.9	11.9	11.9	11.9
Electron affinity (eV)	4.05	4.05	3.8–4.2	3.8–4.2	3.8–4.2
Band gap (eV)	1.12	1.12	1.1–2.2	1.1–2.2	1.1–2.2
Effective conduction band density (cm^−3^)	2.8 × 10^19^	2.8 × 10^19^	1 × 10^20^	1 × 10^20^	1 × 10^20^
Effective valence band density (cm^−3^)	1.04 × 10^19^	1.04 × 10^19^	1 × 10^20^	1 × 10^20^	1 × 10^20^
Electron mobility (cm^2^ V^−1^ s^−1^)	1041	1041	20	20	20
Hole mobility (cm^2^ V^−1^ s^−1^)	418	418	5	5	5
Acceptor concentration (cm^−3^)	2 × 10^17^	0	0	1 × 10^20^	0
Donor concentration (cm^−3^)	0	2 × 10^17^	0	0	1 × 10^20^
Thermal velocity of electrons (cm s^−1^)	1 × 10^7^	1 × 10^7^	1 × 10^7^	1 × 10^7^	1 × 10^7^
Thermal velocity of holes (cm s^−1^)	1 × 10^7^	1 × 10^7^	1 × 10^7^	1 × 10^7^	1 × 10^7^
Layer density (g cm^−3^)	2.328	2.328	2.328	2.328	2.328
Auger recombination coefficient for electron (cm^6^ s^−1^)	2.2 × 10^−31^	2.2 × 10^−31^	0	0	0
Auger recombination coefficient for hole (cm^6^ s^−1^)	9.9 × 10^−32^	9.9 × 10^−32^	0	0	0
Direct band-to-band recombination coefficient (cm^3^ s^−1^)	0	0	0	0	0
Position of oxygen defect	E_V_ + 0.55	E_V_ + 0.55	-	-	-
Density of states (cm^−3^ eV^−1^)	1 × 10^10^	2.5 × 10^9^	-	-	-
σ_e_ (σ_h_) for single defect states	1 × 10^−14^ (1 × 10^−14^)	1 × 10^−14^ (1 × 10^−14^)	-	-	-

**Table 2 materials-15-03508-t002:** Band diagram offset between i-a-Si/c-Si.

**Conduction Band Offset**	χ (eV)				
	3.8	3.9	4	4.1	4.2
ΔE_c_= χ_a−Si_ − χ_c−Si_	−0.25	−0.15	−0.05	0.05	0.15
**Valence band offset**	χ (eV)				
	3.8	3.9	4	4.1	4.2
E_g_(eV)					
1.1	−0.27	−0.17	−0.07	0.03	0.13
1.2	−0.17	−0.07	0.03	0.13	0.23
1.3	−0.07	0.03	0.13	0.23	0.33
1.4	0.03	0.13	0.23	0.33	0.43
1.5	0.13	0.23	0.33	0.43	0.53
1.6	0.23	0.33	0.43	0.53	0.63
1.7	0.33	0.43	0.53	0.63	0.73
1.8	0.43	0.53	0.63	0.73	0.83
1.9	0.53	0.63	0.73	0.83	0.93
2	0.63	0.73	0.83	0.93	1.03
2.1	0.73	0.83	0.93	1.03	1.13
2.2	0.83	0.93	1.03	1.13	1.23

## Data Availability

The data presented in this study are available upon request from the corresponding author.

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
