# Peer review of "A Comparative Study on p- and n-Type Silicon Heterojunction Solar Cells by AFORS-HET"

_materials, 2022, doi:10.3390/ma15103508_

Round 1
Reviewer 1 Report
Review Comments on the Manuscript Materials - 1614207
I think the manuscript "A Comparative Study on p and n-type Silicon Heterojunction Solar Cells by AFORS-HET" by Wabel Mohammed Alkharasani and other authors is an interesting work. It focuses on the comparison between p and n-type Silicon Heterojunction Solar Cells using computer software (AFORS-HET numerical software). Nowadays, a significant contribution of science works in the field of material engineering is related to the development of new material solutions in the area of designing selected functional layers in the technological processes of producing semiconductor heterostructures. I think the quality of the work is suitable for publishing in Materials. Here are some of my suggestions for the authors to consider:
- Editing
- Page 1, line 13- there is “counterpart”, but should be “counterparts”.
- Page 1, line 16- there is ”using n-type”, but should be “using an n-type”.
- Page 1, line 16- there is ”cost-efficiency”, but should be “ a cost-efficiency”.
- Page 1, line 36- there is ”further”, but should be “ for further”.
- Page 2, in the introduction, the order of citation should be followed, e.g. line 68, 70, 72.
- Page 3, line 183 - the board should fit entirely on one side.
- Page 4, line 124- there should be one break line between the table and the text.
- Page 4, line 133 - the period should be removed, there is “Fermi level. [6].”, but should be “Fermi level [6].”
- Page 4, line 141 - the font size in the figures is not legible. This should be corrected.
- Page 5, line 183 - the quality of the figures should be improved and they should be enlarged.
- Page 9, lines 273-274 - figure 5 should fit entirely on the page. It should be reinforced.
- Page 9, lines 275 and 307 - first, there should be a description of the drawing in the article, and then the figure itself (6 and 7).
- Page 11, line 339 - figure 8 should be improved in terms of quality.
- Page 11, line 318 - there is ”Fig.7b-c”, but it should be the beginning of the sentence “ Figures 7b-c”.
- Page 11, lines: 324, 326, 328 - there is “ Fig.3c, Fig.7c, Fig7.a” but should be for an example ’’Fig. 3c”, there is no space between Fig. and the drawing number. Review the text and correct this.
- Pages 12,13, lines 359, 373, 391 - first place the text and then the figure below it. This should be corrected.
- Page 13, line 390 - the figure 11 should be completely on one side.
- Page 13, line 396 - please put "χ" on the line above.
- Pages 13, 14 lines 395, 420, 424, 426 – there is “Eg”, but should be “Eg”.
- Page 14, line 402 – there is “ Fig.11 and 12”, but should be at the beginning of the sentence” Figures 11- 12”.
- Page 14, line 407 – there is “the shallow defects”, please write examples.
- Page 16, lines 467, 470, 517 – there is for example ” June 2019 2019”. This should be corrected.
- Literature should be checked again and where the top index is required, it should be used, for example, line 503 (40 mA / cm2, "), but should be (40 mA/cm2), line 517 there is “ in 2019 27th”, but should be „in 2019 27th”.
- Substantive
- The paragraph on lines 187 to 201 does not detail Figure 4d.
- Page 10 line 316, there is the sentence: ’’the defects have affected the overall cell performance and reduce the active area in structure”. What kind of defects is mentioned in the text? Please explain which ones have a greater or lesser impact on the bandgap values.
- Page 11 line 337, there is the sentence: ’’This could be a result due to a greater loss in the FF for this type of structure [13]”. Please explain that each of these losses may result?
- Page 12 – line 367 - How can you understand deep defects? Please specify it. The work should explain and classify these defects.
Therefore, I recommend that this manuscript consider publication, after taking into account the editing and substantive corrections.
Author Response
Thank you very much for agreeing with us to the intention of this manuscript and helping in improving the manuscript quality. We tried to address all of your suggestions to the best of our knowledge.
Editing
- [Comment 1-4,8-20,22]: Some language and style checked
Response: All the of the changes are made based on the suggestion.
- [Comment 5]: Page 2, in the introduction, the order of citation should be followed, e.g. line 68, 70, 72.
Response: The citations are in order and some references were used in multiple parts.
- [Comment 21]: Page 14, line 407 – there is “the shallow defects”, please write examples.
Response: The example has been added as requested. Page 14, line 412-415.
- [Comment 23]: Literature should be checked again and where the top index is required, it should be used, for example, line 503 (40 mA / cm2, "), but should be (40 mA/cm2), line 517 there is “ in 2019 27th”, but should be „in 2019 27th”.
Response: This happened due to the use of reference program EndNote which insert the name based of the data downloaded from the journal’s website. However, this issue been addressed manually.
Substantive
- [Comment 1]: The paragraph on lines 187 to 201 does not detail Figure 4d.
Response: Although it is mentioned on page 5 that Fig.4d is taking from figure 3, we have added another reminder on page 6, line 212.
- [Comment 2]: Page 10 line 316, there is the sentence: ’’the defects have affected the overall cell performance and reduce the active area in structure”. What kind of defects is mentioned in the text? Please explain which ones have a greater or lesser impact on the bandgap values.
Response: The type of defects is mentioned on page 10 line 315-318. Generally, the type of defects is explained at the beginning of each related section while the overall effect on the bandgap is stated in the conclusion.
- [Comment 3]: Page 11 line 337, there is the sentence: ’’This could be a result due to a greater loss in the FF for this type of structure [13]”. Please explain that each of these losses may result?
Response: The example has been added as requested. Page 11-12, line 346-349.
- [Comment 4]: Page 12 – line 367 - How can you understand deep defects? Please specify it. The work should explain and classify these defects.
Response: The example has been added as requested. Page 13, line 476-478.
Reviewer 2 Report
- The "HIT" abbreviation stands for what ?
- Fig 1. n-type and p-type represent the middle Si layers (160 um). I don't understand how to define your n-i-p and p-i-n structures.
- What's the benefit of using AFORS-HET software ?
Author Response
Thank you for the comments and for helping in improving the manuscript quality. We tried to address all of your suggestions to the best of our knowledge.
- [Comment 1]: The "HIT" abbreviation stands for what?
Response: Heterojunction with an intrinsic thin layer which is also mentioned on page 1 line 35.
- [Comment 2]: Fig 1. n-type and p-type represent the middle Si layers (160 um). I don't understand how to define your n-i-p and p-i-n structures.
Response: The name came from semiconductor electronics after the induction of HIT solar cells as they call the structure based on the dopant layers (160 um layer is the base layer which is the silicon wafer) also can be named based on the emitter location whether it’s front emitter/rear emitter. Please refer to line 104 - 106
- [Comment 3]: What's the benefit of using AFORS-HET software?
Response: AFORS-HET is a software mainly developed for silicon heterojunction studies, more information is presented on Page3, line 100.
Reviewer 3 Report
The article describe a systematic comparison between p and n-type SHJ solar cells was executed in this work by using AFORS-HET numerical software. The article provide good insight to development of solar cell study, especially to reach more efficiency and cost. In order to improve the article quality, here, the several suggestions that can be considered:
- In my opinion, it will be better when the authors add the explanation why the HIT is more important to studied compare to the other concept of solar cell such as perovskite, photonic crystal solar cell and the others. It is important because the author investigate the model of solar cell optimization including their efficiencies. Here, some literatures that can be used for comparison purposes for your study and added to the research motivation part:
https://doi.org/10.1038/s41598-019-48981-w
https://doi.org/10.3390/app12010351
https://doi.org/10.1038/s41586-021-03285-w
- The author state that 4 HIT established silicon solar cell structure considered from the literature (line 92). However, there is no citation in this part.
Author Response
[General Comment]: the article describe a systematic comparison between p and n-type SHJ solar cells was executed in this work by using AFORS-HET numerical software. The article provide good insight to development of solar cell study, especially to reach more efficiency and cost.
Response: Thank you for the comments and for helping in improving the manuscript quality. We tried to address all of your suggestions to the best of our knowledge.
In order to improve the article quality, here, the several suggestions that can be considered:
- [Comment 1]: In my opinion, it will be better when the authors add the explanation why the HIT is more important to studied compare to the other concept of solar cell such as perovskite, photonic crystal solar cell and the others. It is important because the author investigate the model of solar cell optimization including their efficiencies. Here, some literatures that can be used for comparison purposes for your study and added to the research motivation part:
Response: We have added the comparison in the introduction and cited the recommended references. Line 78-88
- [Comment 2]: The author state that 4 HIT established silicon solar cell structure considered from the literature (line 92). However, there is no citation in this part.
Response: We have added the related references as requested. Ref 13 & 16 in line 104.